# Network Pharmacology and Molecular Docking Elucidate the Underlying Pharmacological Mechanisms of the Herb *Houttuynia cordata* in Treating Pneumonia Caused by SARS-CoV-2

**DOI:** 10.3390/v14071588

**Published:** 2022-07-21

**Authors:** Junying Liu, Shouli Yuan, Yao Yao, Jinfan Wang, Gaia Scalabrino, Shibo Jiang, Helen Sheridan

**Affiliations:** 1NatPro Center, School of Pharmacy and Pharmaceutical Sciences, Trinity College Dublin, D02PN40 Dublin, Ireland; juliu@tcd.ie (J.L.); wangj5@tcd.ie (J.W.); scalabrg@tcd.ie (G.S.); 2Academy for Advanced Interdisciplinary Studies, Peking University, Beijing 100080, China; yuanshouli123@pku.edu.cn; 3Biocomputing and Developmental Systems, Lero—The Science Foundation Ireland Research Centre for Software, Department of Computer Science & Information Systems, The University of Limerick, V94T9PX Limerick, Ireland; yao.yao@ul.ie; 4Key Laboratory of Medical Molecular Virology (MOE/NHC/CAMS), School of Basic Medical Sciences, Shanghai Institute of Infectious Disease and Biosecurity, Fudan University, Shanghai 200032, China

**Keywords:** *Houttuynia cordata*, network pharmacology, pneumonia, SARS-CoV-2, molecular docking, afzelin, MAPK1

## Abstract

Used in Asian countries, including China, Japan, and Thailand, *Houttuynia cordata* Thumb (*H. cordata*; Saururaceae, HC) is a traditional herbal medicine that possesses favorable antiviral properties. As a potent folk therapy used to treat pulmonary infections, further research is required to fully elucidate the mechanisms of its pharmacological activities and explore its therapeutic potential for treating pneumonia caused by SARS-CoV-2. This study explores the pharmacological mechanism of HC on pneumonia using a network pharmacological approach combined with reprocessing expression profiling by high-throughput sequencing to demonstrate the therapeutic mechanisms of HC for treating pneumonia at a systemic level. The integration of these analyses suggested that target factors are involved in four signaling pathways, including PI3K-Akt, Jak-STAT, MAPK, and NF-kB. Molecular docking and molecular dynamics simulation were applied to verify these results, indicating a stable combination between four metabolites (Afzelin, Apigenin, Kaempferol, Quercetin) and six targets (DPP4, ELANE, HSP90AA1, IL6, MAPK1, SERPINE1). These natural metabolites have also been reported to bind with ACE2 and 3CLpro of SARS-CoV-2, respectively. The data suggest that HC exerts collective therapeutic effects against pneumonia caused by SARS-CoV-2 and provides a theoretical basis for further study of the active drug-like ingredients and mechanism of HC in treating pneumonia.

## 1. Introduction

*Houttuynia cordata* Thumb (*H. cordata*; Saururaceae) is a traditional herbal medicine used in Asian countries, including China, Japan, and Thailand. It exhibits promising antiviral activities towards clinically enveloped viruses, such as influenza virus, herpes simplex virus-1(HSV-1), and human immunodeficiency virus-1 (HIV-1) in vitro [1]. As a time-honored traditional Chinese medicine (TCM), HC has demonstrated a broad range of pharmacological activities for the treatment of inflammatory diseases, especially pulmonary symptoms, i.e., phlegm, dyspnea, lung abscess, and cough, and such infectious diseases as severe acute respiratory syndrome (SARS) [2]. This herb has traditionally been used as one of six principal ingredients in an herbal formula purported to have a preventive effect on SARS-CoV-1 infection [3,4]. As such, it has been recommended to the general public as a preventative measure by the State Administration of TCM of China for SARS-CoV-2 [5]. As a potent folk treatment of pulmonary infections, further research is required to fully elucidate the mechanisms of its pharmacological activities and explore its therapeutic potential for treating lung inflammatory disorders, particularly pneumonia caused by SARS-CoV-2.

HC consists of a large number of drug-like bioactive compounds that have been isolated in previous studies and could play a vital role in antiviral and anti-inflammatory medicine [6,7], namely, volatile oil, alkaloid, flavonoid, polysaccharides, and organic acid. The main volatile oils of methyl-n-nonyiketone, decanoylacetadehyde, β-myrcene, and α-pinene, extracted by steam distillation, demonstrated a direct antiviral effect on HSV-1, influenza virus, and HIV-1 [8]. The main water extracts of HC with quercetin, quercitrin, and Quercetin–3-glucoside can also prevent HSV-1 infection by inhibiting the activation of NF-kB [9], while the polysaccharide can inhibit inflammation, protect the intestinal barrier, and regulate mucosal immunity to alleviate lung and intestinal injuries caused by IAV virus [10]. A recent systematic evaluation of the natural metabolites that could potentially be used against SARS-CoV-2 demonstrated that apigenin, afzelin, kaempferol, quercetin, and quercetin 3-glucosyl-(1,4)-rhamnoside could bind with 3C-like protease (3CLpro) and angiotensin I converting enzyme 2 (ACE2), respectively, at significant binding energy [11]. As previously mentioned, research has primarily focused on various solvent HC extracts’ antiviral and anti-inflammatory mechanisms. Yet, the underlying connection between these bioactivities and HC metabolites remains obscure.

Based on the potential antiviral and anti-inflammatory properties of HC multi-compounds, HC has been used to treat different types of pneumonia caused by bacteria and viruses [12]. Coronavirus disease-19 (COVID-19) can cause lung complications, including pneumonia, and in the worst cases, acute respiratory distress syndrome (ARDS) [13]. Sepsis, another possible complication of COVID-19, can also cause lasting harm to the lungs and other organs [14]. The mechanism underlying the inhibition of HC water extract on viral pneumonia arguably rests on the direct inhibition of the virus by flavonoids and regulation of immune function. At the same time, its polysaccharide reduces cytokines, including tumor necrosis factor-α (TNF-α), interleukin 1 (IL1), and interleukin 6 (IL6), thereby inhibiting innate immune cells and epithelial cells from releasing excessive cytokines, while increasing, or rescuing, the expression of anti-inflammatory mediators like interleukin10 (IL10) and interleukin 14 (IL4) [15]. Based on this hypothesis, this study examines the pharmacological mechanism of HC on pneumonia caused by SARS-CoV-2, using network pharmacology and molecular docking, thus providing a new strategy for the development of novel drugs targeting pneumonia caused by SARS-CoV-2 (Figure 1).

Acute lung injury (ALI) was found to cause pneumonia, matrix lesions, and pulmonary fibrosis initiated by SARS-CoV-1 (2003), avian influenza (2008), swine influenza (2009), and SARS-CoV-2 (2020) [2]. Indeed, ALI is a potentially lethal factor contributing to the short-term acute exacerbation of idiopathic pulmonary fibrosis in ARDS with rapid pulmonary fibrosis [6]. Both SARS-CoV and Middle East respiratory syndrome coronavirus (MERS-CoV) causes severe atypical pneumonia in humans and present with similar symptoms, which contribute to preferential viral replication in the lower respiratory tract and viral immunopathology [16], while pathogenic T cells and inflammatory monocytes incite an inflammatory storm, or cytokine storm, in severe COVID-19 patients [17]. The lung injury pattern of severe COVID-19 patients is similar to ARDS. Characterized by severe, often refractory, hypoxemia and bilateral lung infiltrates, ARDS has been reported in 42% of patients hospitalized in Wuhan with COVID-19 pneumonia [18]). Thus, following an initial hyper-inflammatory phase, many severe COVID-19 patients eventually develop some degree of immune paralysis and increased risk of secondary infections [19], as well as evidence of pulmonary fibrosis.

SARS-CoV-2 infection is a critical factor in the onset of pneumonia and death. HC is a time-honored TCM herb widely used to treat both bacterial and viral respiratory diseases, and it was purported to prevent and treat SARS-CoV-1 infection in 2003. In light of this, effective herbal compounds for SARS-CoV-1 were posited to be beneficial in the treatment of COVID-19 caused by SARS-CoV-2 since homology between the above two viruses is around 80% [5]. As a folk therapy to treat pulmonary infections, HC is one of the major ingredients in the capsule Lian Hua Qing Wen, a TCM formula with the ability to reduce inflammation, which was endorsed by the State Administration of TCM of China [20] as a public treatment measure.

## 2. Materials and Methods

### 2.1. Bioactive Compound Screening and Pharmacokinetic Prediction

HC was searched to identify its active ingredients and pharmacological targets using an in silico approach. As a result, the efficacy of active compounds of HC prediction was established in public databases, including the Traditional Chinese Medicine Systems Pharmacology Database and Analysis Platform (TCMSP), the TCM Integrated Database (TCMID), The Encyclopaedia of Traditional Chinese Medicine (ETCM), and the Bioinformatics Analysis Tool for Molecular Mechanism of TCM (BATMAN-TCM) [21]. In addition, absorption, distribution, metabolism, and excretion (ADME) was also employed as a computational evaluation model in pharmacokinetic research to select drug-like compounds [22]. This model consists of criteria such as drug-likeness (DL) and oral bioavailability (OB). These two indices were applied to ascertain whether the compounds have drug-like properties as therapeutic agents and are chemically suitable for drug development [23]. Out of 50 compounds shown in these databases, compounds with DL > 0.18 and OB > 30% were selected. In total, nine bioactive components were ultimately included in this study and used for the subsequent prediction of compound-related targets.

### 2.2. Potential Targets of HC Active Components

The active components of a drug interact with respective targets to inhibit their biological function. An HC target gene set was acquired by searching several databases: (1) acquiring gene symbols and related information about HC targets from TCMSP; (2) importing selected candidate components into the PubChem database (https://pubchem.ncbi.nlm.nih.gov/ (accessed on 1 August 2021)) to identify relevant targets, and (3) using the ETCM to acquire the target genes associated with the selected active compounds with a score >0.8. The target set was derived after combining the search results and removing duplication and certain bioactive components with suitable targets with a score >0.8 [24].

### 2.3. Identification of Pneumonia-Related Targets Database

Genes related to pneumonia caused by SARS-CoV-2 were screened, selected, and obtained from the Gene Expression Omnibus (GEO), GeneCards database, and other references. First, the dataset was mainly derived from reprocessing high-throughput sequencing data downloaded from the GEO database. The expression profile of GSE152075 involved 430 SARS-CoV-2 infection samples and 54 negative control samples, which were analyzed on the GPL18573 Illumina NexSeq500 platform (Homo sapiens) [25]. The limma package in R Bioconductor was used to identify differentially expressed genes (DEGs) between SARS-CoV-2 infection and negative control, in which the adjusted *p*-value and [logFC] were determined [26]. The selected criteria of DEGs were thereafter set as *p* < 0.05 and [logFC] > 1.08 for up-regulated genes and [logFC] < 1.651 for downregulated genes [25]. Second, the targets related to viral pneumonia were obtained by using “Viral pneumonia” as the keyword in the GeneCards database search (https://www.genecards.org/ (accessed on 10 August 2021)). Finally, an intersection between genes retrieved through GeneCards and DEGs of reprocessed data series GSE152075 was obtained as pneumonia-related targets [27].

### 2.4. Construction of PPI Network and Herb-Metabolites-Targets-Disease (HMTD) Network

The HC targets intersecting with pneumonia-related targets were taken as HC–pneumonia common targets, which can be visualized with Venn 2.0. The common targets were imported into the STRING platform (version 11.0). The species was then set to Homo sapience and the minimum required interaction score to the highest confidence of 0.9 in order to retrieve the concise protein–protein interaction (PPI) information for the next step of the analysis. The PPI network was also visualized with Cytoscape software [28].

The Herb-Metabolites-Targets-Disease (HMTD) network was built on the interactions among drug (HC), ingredients, gene symbols, and disease (pneumonia) and then visualized by Cytoscape software. The nodes’ varying shapes represent common pneumonia and HC active ingredient targets. The nodes are linked by edges (lines), indicating interactions between nodes. As such, the topological features of a network can be used to predict the targets, while the candidate hub notes analyzed by Cytoscape’s Network analyzer tool can be identified by calculating the two topological features of betweenness and degree. Betweenness is the number of shortest paths through the node (the shortest distance between two nodes) and degree is the sum of the number of edges connected the node [29]. Closeness centrality measures how close a node is to all others in the same network [30]. These network centrality indices have been used to define the network properties of drug targets separately or collectively and judge the importance of nodes. Nodes (targets) with higher ranks were considered to have a more critical role within the network [31]. Collectively, the top 15 nodes were screened as the hub genes in the network with the criteria of closeness centrality >0.4 and degree > 4.

### 2.5. GO and KEGG Analysis

Depending on the hub genes (core targets), the Gene Ontology (GO) biological processes, and the Kyoto Encyclopaedia of Genes and Genomes (KEGG), metabolic pathway enrichment analyses were carried out on the pneumonia–HC common targets. GO source and GO enrichment can divide the functions and products of genes into three categories, namely, molecular function (MF), biological process (BP), and cellular components [32]. The biological processes and pathways selected from the analysis of Metascape were colored by cluster ID with the best *p*-values from each of 20 clusters, wherein nodes that share the same cluster ID are typically close to each other. Enrichment analysis was also carried out in PaGenBase to demonstrate disease targets–organs location. In addition, a further enrichment analysis was performed in DisGeNET to identify the relevant diseases. The results of the KEGG enrichment analysis were used to construct a KEGG pathway network to determine the proteins involved in the treatment effects of HC. Based on the STRING results, the gene–pathway of HC against pneumonia was constructed to delineate the various pathways and key targets in order to explore the potential mechanisms underlying the effect of HC on the treatment of pneumonia [28].

### 2.6. Molecular Docking

Molecular docking is a useful tool to predict and design new drugs. As such, the computational validation of ingredients–targets interactions were confirmed by exploring their binding modes via this process. The computational modeling of intermolecular combinational patterns between target proteins and herb ligands can predict the potential binding modes. Depending on the degree of 67 common targets in the PPI network and the important reference, four active ingredients, including afzelin, apigenin, kaempferol, and quercetin, and six targets, i.e., dipeptidyl peptidase-4 (DPP4), neutrophil elastase (ELANE), Heat shock protein (HSP90AA1), IL6, mitogen-activated protein kinase 1 (MAPK1), and Serpin Family E Member 1 (SERPINE1), were selected to simulate the ingredients–targets interactions for verification of molecular docking. Molecular 2D structures for the molecular ligands and active ingredients were downloaded from PubChem databases. The crystal structures of the key target proteins DPP4 (PDB ID: 4N8D), ELANE (PDB ID: 4WVP), HSP90AA1 (PDB ID: 5J2X), IL6 (PDB ID: 4CNI), MAPK1 (PDB ID: 6QAH), and SERPINE1 (PDB ID: 7AQF) were selected from RCSB PDB (https://www.wwpdb.org/ (accessed on 11 August 2021)). The key target proteins were purposefully selected with a resolution smaller than 2, and their crystals were imported into PyMOL 3.0 software [33]. The active site of the protein is centered on the active amino acid site of the original ligand in the crystal structure, which residue information can be obtained from the literature [34,35,36,37,38,39].

Docking was performed by Autodock Vina 1.1.2, and the molecules with the lowest binding energy in the docking conformation were chosen to observe the binding effect by matching with the original ligands and intermolecular interactions, such as hydrophobic interaction, cation–π, hydrogen bond, anion–π, π–π stacking, salt bridge, and metal complexation [33]. The molecular docking patterns were finally visualized via PyMOL 3.0.

### 2.7. Molecular Dynamics Simulation

The molecular dynamics simulations of these complexes were performed using Gromacs 2020.1, in which the charm36-jul2020 force field was chosen. The complex was solved in TIP3P water and immersed in a dodecahedron box extending to at least 1 nm of the solvent on all sides. The system was neutralized by Na^+^ and Cl^−^, followed by adding 0.15 M NaCl, which made the system close to the physiological state. The system was minimized by using the steepest descent algorithm for 5000 steps and made a maximum force of less than 1000 kJ/mol/nm. Then, it was equilibrated in a constrained NVT (number of particles, volume, temperature) and NPT (number of particles, pressure, temperature) running for 100 ps. The system was well-equilibrated through NVT and NPT equilibration at 300 K and 1 bar. Finally, MD simulations of the complex were carried out for 100 ns. The Verlet cut-off scheme and a Leap-frog integrator with a step size of 2 fs were applied. The final analysis of molecular dynamics included the root mean square deviation (RMSD) of protein and molecule and the interaction energy between the protein and small molecules, which were calculated by GROMACS 2020.1.

## 3. Results

### 3.1. Target Prediction and Analysis of HC

While 50 metabolites were shown in the TCMSP by searching the keywords “Houttuyniae Herba”, only 7 satisfied the criteria of OB ≥ 30% and DL ≥ 0.18. Other metabolites were obtained by searching ETCM, BATMAN-TCM, and TCMID. A number of metabolites had already been established as the most effective components of HC throughout the relevant literature [40]. These were also included, even though they did not meet the OB and DL criteria. As such, a total of 20 metabolites were acquired. Depending on these metabolites, the targets for a number of active ingredients of HC were identified by target fishing and by integrating the data acquired from TCMSP, PubChem, and ETCM. The targets of each active ingredient derived from the ETCM database were selected via the screening score ≥0.8. Therefore, only 7 active metabolites were left after searching the targets in those databases, including quercetin, quercitrin, kaempferol, acetyl borneol, decanoic acid, afzelin, and apigenin (Table 1), and 463 targets related to the above seven core active metabolites were identified.

### 3.2. Disease Targets Analysis

Bioinformatics analyses on the expression profile microarray data GSE152075 and GSE1739 containing positive SARS-CoV-2 and negative control samples were performed to identify DEGs between SARS-CoV-2 infection and negative control by the limma package in R Bioconductor. This step identified 9685 DEGs from data series GSE152075 and 1791 DEGs from data series GSE1739. Other disease data sources, such as GeneCards and DisGeNET, were combined with the GEO results to remove duplicates, resulting in the identification of 11,027 targets related to SARS-CoV-2. Since SARS-CoV-2 causes not only pneumonia but also multiple organ failure, neutrophilia, and organ and coagulation dysfunction, pneumonia-related targets were acquired by searching GeneCards with the keywords “SARS-CoV-2” and “pneumonia”, resulting in 786 pneumonia-related targets. The intersection between SARS-CoV-2-related targets (11,207) and pneumonia-related targets (786) resulted in 739 elite targets related to pneumonia caused by SARS-CoV-2.

### 3.3. Herb-Ingredients-Targets-Disease Network of HC Analysis

The intersection between HC-related targets (463) and pneumonia-related elite targets (739) resulted in an HC–pneumonia common target set with 67 genes. This common target set was imported into Cytoscape v3.5.0 to construct an Herb-Metabolites-Targets-Disease (HMTD) network, as shown in Figure 2. This consisted of six metabolites assigned to 67 targets, indicating HMTD interactions. The node can be designed as a hub node if the degree, betweenness, and closeness satisfy specific criteria, such as the median of the corresponding parameters. The screening of important metabolites and core targets was carried out based on the criteria of SUID > 70, Closeness Centrality > 0.4 and Degree > 4, resulting in four ingredients (apigenin, quercetin, afzelin, and kaempferol) and 21 core common targets (Table 2). Consequently, these important metabolites might be crucial active compounds of HC targeting 21 genes, which could be verified through molecular docking or further experiments. The HC–pneumonia common target set was imported into STRING to remove unconnected targets, and a PPI network with a confidence score set to 0.9 or higher was gained. PPI information from the STRING platform was input to Cytoscape software to construct a PPI network based on the common targets shown in Figure 3. The size of target nodes was consistent with the degree, and the nodes with pink color were deemed to be important targets [41]. According to the degree and combined score, the top 21 common targets shown in Table 2 are involved in the effects of HC treatment on pneumonia.

### 3.4. GO and KEGG Enrichment Analysis

According to the results of GO enrichment analyses in Metascape, the genes were enriched in different GO terms, and the top 20 GO terms in the three categories were selected to construct connections within the signaling network (Figure 4). The network was colored by cluster ID with the best *p*-values from each of 20 clusters, wherein nodes that share the same cluster-ID are typically close. The top signaling pathways mainly include interleukin-4 and interleukin-13 signaling, the AGE–RAGE signaling pathway in diabetic complication, positive regulation of cell migration, spinal cord injury, IL17 signaling pathway, reactive oxygen species metabolic process, leukocyte activation involved in immune response, Th17 cell differentiation, cytokines and inflammatory response, and epithelial cell migration (Figure 4). The x-axis in the bar chart represents log 10 (*p*-value) and the y-axis the GO term. The enrichment analysis in PaGenBase demonstrated disease targets-organs location, such as lung, smooth muscle, cardiac myocytes, bone marrow, bronchial epithelial cells, liver cells, and spleen. The analyses of the most-associated diseases showed immunosuppression, fatty acid disease, respiratory distress syndrome, pneumonitis, endothelial dysfunction, respiratory syncytial virus infection, liver failure, middle cerebral artery occlusion, bacterial infections, acute myocardial infarction, lung diseases, cardiac arrest, myocardial ischemia, and herpes simplex infections. To analyze the significance and importance of key targets in the pathways involved in the treatment effect of pneumonia, 10 key pathways, determined according to gene counts and adjusted *p* values from the KEGG enrichment analysis and related targets, were used to construct a KEGG key pathway network (Figure 5). The construction of gene–KEGG key pathways demonstrated that the targets MAPK1, MAPK3, IL6, PIK3CA, AKT1, EGFR, TNF, and STAT3 involved more than four signaling pathways, including PI3K-Akt, Jak-STAT, MAPK, and NF-kB (Table 3). As such, HC could target multiple functional and biological factors in pneumonia. However, the effects and profound influence required further validation [28].

### 3.5. Docking and Molecular Dynamics Simulation Analysis of Ingredients-Targets

Considering the integration of the results from the PPI network and HMTD network, the key targets in the pathways mentioned above and the nodes with high degrees represent the key targets. Therefore, molecular docking validation was performed based on the pneumonia-related targets and selected ingredients from the HMTD network. The selected active ingredients included quercetin (CAS no. 117-39-5), kaempferol (CAS no. 520-18-3), afzelin (CAS no. 482-39-3), and apigenin (CAS no. 520-36-5). The protein structures of key targets were acquired online from RCSB PDB, including DPP4, ELANE, HSP90AA1, IL6, MARK1, and SERPINE1, based on STRING interaction analysis and importance reference. Docking analysis of the metabolites and proteins above showed the docking patterns and binding affinities (Table 4). The docking results were represented on the molecular surface to reflect the topical details of the binding sites. The residues were marked on the protein surface, and hydrogen bonds were shown as solid lines (Figure 6). The binding affinities of all docking patterns were less than −6 kcal/mol, indicating a stable binding between active ingredients and protein targets. The binding affinities are listed in Table 4, and the binding configuration is shown in Figure 6. The affinity energy of the best mode, apigenin, is −9.4 kcal/mol.

In order to verify the stability of the docking structures, we selected DPP4-–kaempferol, MAPK1–afzelin, SERPINE1–apigenin and SERPINE1–quercetin complexes for dynamic simulation analysis. As shown in Figure 7, RMSD of proteins and small molecules in the complex structures remained relatively stable during the simulation, especially MAPK1–afzelin, SERPINE1–apigenin and SERPINE1–quercetin complexes. The RMSD of kaempferol in DPP4–kaempferol varied greatly, and it was stable at 1 nm compared with the initial docking structure, indicating that the position of kaempferol changed significantly during the simulation process. This position change occurred rapidly and remained stable at the new binding position. The average interaction energy of DPP4–kaempferol, MAPK1–afzelin, SERPINE1–apigenin and SERPINE1–quercetin complexes was −107.34 kJ/mol, −180.39 kJ/mol, −175.82 kJ/mol, and −183.77 kJ/mol.

In conclusion, we used docking and dynamics simulation to explore ingredient-target prediction. Our results showed that the DPP4–kaempferol, MAPK1–afzelin, SERPINE1–apigenin, and SERPINE1–quercetin complexes had good docking fractions, that the protein and small-molecule positions were stable during the simulation process, and that the interaction energy was lower than −100 kJ/mol. Accordingly, our studies provide a reference for subsequent experimental design.

## 4. Discussion

While previous studies demonstrated that HC exhibited antiviral activity through inhibiting SARS-CoV-1 3CLPRO and RdRp and also stimulated the proliferation of CD4^+^ and CD8^+^ T cells in vitro [3], the ability of HC to inhibit SARS-CoV-2 infection and the similarities between the inhibitory mechanisms of SARS-CoV-2 and SARS-CoV-1 remain unknown. Although several clinical studies have confirmed that prescriptions or formulas (Lian Hua Qing Wen capsule) containing HC are effective for the treatment of COVID-19 [42], further research is necessary to investigate the mechanism of action of HC and apply the pharmacology of TCM network for the predictive analysis. Thus, this study aimed to explore the pharmacological mechanism of HC on pneumonia caused by SARS-CoV-2 using network pharmacology and molecular docking.

TCM considers an individual or patient an integrative complex with dynamic states, demonstrating multiple biological targets and focusing on integral therapeutic efficacies [43]. Inflammation has been the pathophysiological mechanism behind many chronic diseases, including cytokines, nitric oxide (NO), lipid mediators, G prostaglandins, and leukotrienes produced by macrophages, neutrophils, and other inflammatory cells [44,45]. This study included seven core metabolites, following diligent screening and searching in the references, including quercetin, quercitrin, kaempferol, acetyl borneol, decanoic acid, afzelin, and apigenin. These flavonoids are known to be large entities of plant constituents and possess anti-inflammatory activity [15]. Some flavonoids have been shown to attenuate lung inflammatory response strongly. For example, quercetin was previously reported to attenuate lipopolysaccharide (LPS)-induced lung inflammation in mice by oral administration [14], while afzelin isolated from methanol extract of HC was demonstrated to regulate both mitophagy and mitochondrial biogenesis through Rev–Erb-/phosphor–AMPK/SIRT1 signaling [40]. In addition, afzelin can inhibit mitochondrial dysfunction induced by excessive oxidative stress and attenuate the reduction of mitochondrial GDH activity and hepatic ATP production in LPS-induced hepatic injury [40]. Kaempferol has been proven to protect against H9N2 swine influenza virus infection and can ameliorate virus-induced acute lung injury by inactivation of TLR4/MyD88-mediated NF–kB and MAPK signaling pathways [46]. Kaempferol can also inhibit the release of TNF-α, IL1β, IL6, and IL18 and suppress the activation of NF–kB and AKT, thus attenuating cardiac fibroblast inflammation [47]. Apigenin, a plant-derived flavonoid, possesses anti-carcinogenic, antioxidant, anti-inflammatory, and anti-mutagenic properties [48]. Apigenin can react to the Nrf2 gene, which encodes a key transcription factor to regulate the antioxidative defense system, and was also a potent inhibitor of SARS-CoV 3CLpro [47]. The targets of these active ingredients were collected from a different database.

Depending on the above active metabolites, target searching in the databases of TCMSP, PubChem, and ETCM resulted in the acquisition of 463 targets related to the above seven core active metabolites. The disease targets were obtained by identifying key genes in SARS-CoV-2 infection and uncovering their potential functions by re-processing the expression profiling of high-throughput sequencing of GSE152075 from the GEO database to form a solid basis for the ensuing analysis. The bioinformatics analyses generated 11,207 DEGs, contributing to our understanding of the molecular mechanism underlying the advancement of SARS-CoV-2 infection.

Since SARS-CoV-2 causes multiple organ failure [49], the intersection between DEGS acquired from the GEO series and pneumonia-related genes from GeneCards and DisGeNET isolated pneumonia caused by SARS-CoV-2, resulting in 739 elite targets. These were matched and mapped to obtain 67 common HC-pneumonia targets. A PPI network of the common targets and screened nodes according to observed gene count > 50 resulted in 21 core target networks to verify targets associated with HC ingredients. According to the combined score of each node, the top 25 nodes were mainly NOS3, MTOR, SERPINE1, PIK3CA, HSP90AA1, STAT3, INS, IL6, IL-1β, TNF, IL10, VEGFA, CDK1, and GSK3B. Most of these genes are associated with inflammation, hypoxia, and angiogenesis [50]. This result aligned with the findings of previous studies of HC extract and confirmed that HC can down-regulate specific inflammatory mediators, such as TNF-α, IL6, prostaglandin E2 (PGE2), and nitric oxide (NO) production in the cells, inducible nitric oxide synthase (iNOS), and cyclooxygenase-2 (COX-2) expression [7].

The pathogenesis of inflammatory diseases is associated with the overproduction of the above mediators. NO by endothelial nitric oxide synthase (NOS3) is implicated in vascular smooth muscle relaxation and mediates vascular endothelial growth factor (VEGF)-induced angiogenesis in coronary vessels, which may explain the many complications occurring in COVID-19 patients [51]. The mammalian target of rapamycin (mTOR) entails the downstream of PI3K–Akt to regulate cell growth and proliferation, cell survival, protein synthesis, and transcription [52]. HC may inhibit PI3K/Akt/mTOR and ERK1/2 signaling pathways in human lung cancer cells [53], while IL17A can inhibit PI3K/Akt/mTOR-mediated autophagy, which causes lung inflammation and fibrosis [54]. However, alternative evidence suggests that activation of the PI3K/Akt/mTOR pathway may contribute to pulmonary fibrosis and lung injury by regulating lung fibroblasts and epithelial cells [55]. For STAT3, the activation of JAK–STAT signaling can lead to fibrosis in many organs, i.e., the lung [47]. STAT3 is a potential molecular target for clinical syndromes characterized by systematic inflammation in COVID-19 in a large-scale transcriptional study [56]. Therefore, GO and KEGG enrichment analysis of the core targets was applied to explore the underlying mechanism of HC.

The GO enrichment results confirmed that HC mainly regulates biological processes in response to stimulus, regulation of the cellular process, response to chemicals, response to stress, cell communication, signal transduction regulation of macromolecule metabolic process, and regulation of nitrogen compound metabolic processes. The results of KEGG enrichment uncovered the numerous signaling pathways involved in the development and progression of pneumonia, including PI3K-Akt, HIF-1, IL-17, TNF, TLR, JAK–STAT, NOD-like receptor, or MAPK. Previous studies further suggest that the anti-inflammatory properties of HC extract may arise from the inhibition of pro-inflammatory mediators by suppression of NF–κB, and MAPK signaling pathways by binding their key proteins in pathways with HC active metabolites, as outlined above [9,39,41].

Based on the PPI, HMTD, and KEGG enrichment analysis, 6 of 21 core targets, namely, DPP4, ELANE, HSP90AA1, IL6, MAPK1, and SERPINE1, were selected to dock with four main ingredients identified in a series of analyses of this study. As such, the docking results agreed with the intermolecular interactions. Despite extensive prior efforts to elucidate the metabolism and effects of oral administration of HC in patients in vivo, the mechanism remained largely unclear. DPP4, HSP90AA1, and SERPINE1 are related to immune response and linked to macrophages which promote the generation of inflammatory factors, such as TNF-α and IL6 [57]. DPP4 is a member of serine peptidases known as adenosine deaminase complexing protein 2 clusters of differentiation 26 (CD26) associated with immune regulation, apoptosis, and signal transduction. However, the main receptor of SARS-CoV is angiotensin-converting enzyme 2 (ACE2) or CD209L, whereas MERS-CoV uses DPP4 (also known as CD26) as the major receptor [58]. A correlation between DPP4 and ACE2 found that both membrane proteins can facilitate virus entry. Therefore, DPP4 was speculated to be a co-receptor to facilitate SARS-CoV-2 infection since DPP4 can be found in lung cells [59]. The co-receptors of ACE2 and DPP4 to the spike glycoprotein postulate that different human coronaviruses target similar cell types across different human organs [60]. Heat shock proteins (HSPs), also known as stress proteins, are divided into the HSP70 family, HSP90 family, HSP 100 family, and so on. HSP90AA1 is a molecular chaperone protein with inhibitors that can induce hepatic stellate cell apoptosis through neurophospholipase or NK-κB, depending on the mechanism. A previous study found that Jin Hua Qing Gan Granules regulate multiple signaling pathways via binding targets, such as PTGS2, HSP90AA1, and NCOA2, to prevent COVID-19 [61].

ELANE genes can express up to a five-fold level against SARS-CoV-1 infection, causing lung proinflammatory cytokines [62]. Recent research found strong evidence that the ELANE gene mainly enhances proinflammatory cytokines and can subsequently cause epithelial cell injuries in cystic fibrosis patients [62,63]. The suppression of the ELANE gene can therefore reduce the production of proinflammatory cytokines, resulting in improved pulmonary function [63]. As such, the inhibition of ELANE directly protects the lung and reduces lung inflammatory cell infiltration to improve the success rate of coronavirus patients [62]. MAPK1 is the critical target participating in the MAPK signaling and PI3K-Akt signaling pathways. The inhibition of MAPK1 can result in inhibiting the above two pathways in LPS-induced ALI in mice [64]. SERPINE1 is a serpin peptidase inhibitor whose increased expression resulted in a lower survival rate. Furthermore, increased expression of SERPINE1 was associated with the activation of the PI3K–Akt pathway [16]. The results from PPI and HMTD indicated that the six intersection genes above could prove to be potent pharmacological targets of HC against COVID-19.

Molecular docking of 6 out of 21 core targets and four screened active ingredients was applied to validate the results of the network pharmacology. The active ingredients which target these proteins include quercetin, kaempferol, afzelin, and apigenin. Docking within the docking pockets between active ingredients and target proteins was visualized by AutoDock Vina and PyMOL software. The binding affinities of the docking results ranged from −6.7 to −9.4 kcal/mol, indicating stable binding. Afzelin showed the greatest binding affinities with MAPK1 (−9.4 kcal/mol), followed by DPP4 (−8.6 kcal/mol). In contrast, MAPK showed a better average binding affinity with these four active ingredients, indicating that HC protected against acute lung injury, mainly through the suppression of the MAPK/NF–kB pathway [5]. The docking pose of apigenin shows H-bonds between the aromatic region and residues PHE-129, ILE-133, ASN-82, ASP-106, ILE-84, ASN-158, and THR-190, establishing a stacking interaction with GLN-132 [65]. However, the docking results reflect possible treatment mechanisms and guide herb–disease validation via cells and animal experiments.

Network pharmacology has been widely used for TCM mechanism research owing to its efficacy in analyzing the complicated relationships among multiple ingredients and multiple disease targets. Docking technology can visualized the binding modes of active ingredients with disease-related key target proteins and provide guidance for researchers’ selections of active ingredients for in vivo or in vitro experiments [66]. As shown in the binding affinity result, multiple ingredients can bind the same protein, which may result in synergistic effects. This represents a significant challenge for current network pharmacology. In addition, further animal and cell models are needed to verify the relevant pathways and targets.

This study confirmed the potent therapeutic effect of HC, a time-honored herb widely used in Asian countries to treat pneumonia. The potential mechanisms of HC were revealed by employing both network pharmacology and molecular computational analyses. Our results offer a very different perspective in terms of modern pharmacological mechanisms which may assist in the global fight against the COVID-19 pandemic. However, while HC is an effective herb for pneumonia treatment, the optimal dose for inducing remission with low toxicity needs to be determined. Moreover, further animal and cell models are necessary to verify the relevant pathways and targets.

## Figures and Tables

**Figure 1 viruses-14-01588-f001:**
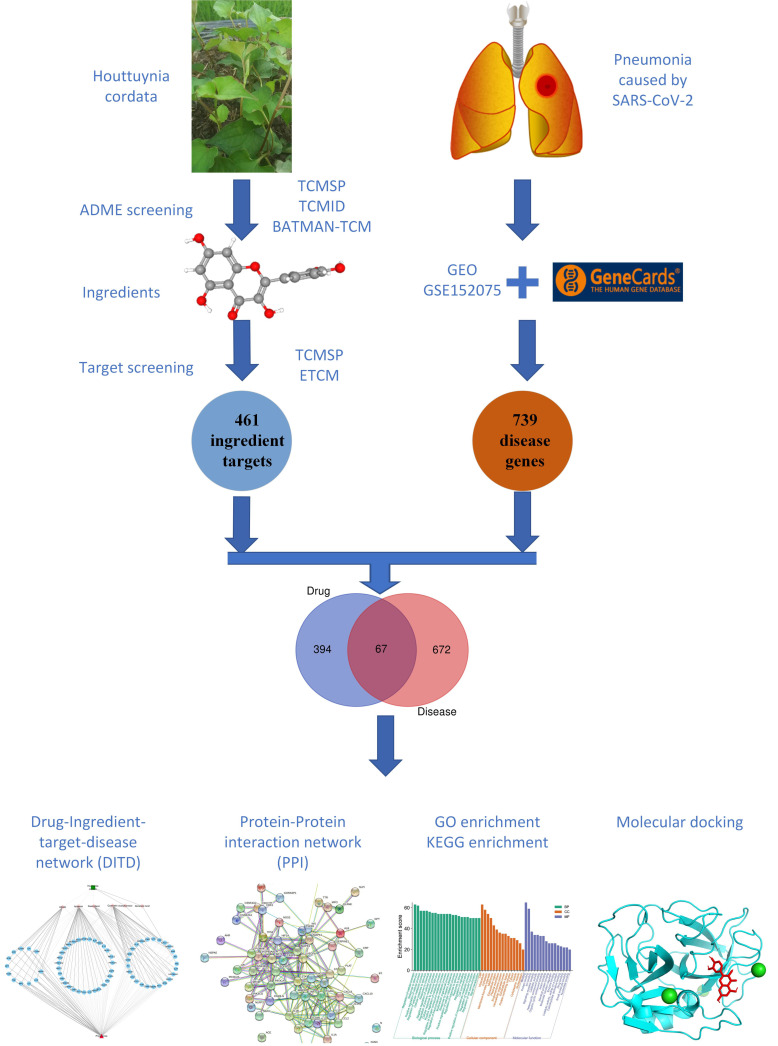
Framework based on an integrative strategy of network pharmacology to investigate pharmacologic mechanisms of *Houttuynia cordata* for the treatment of pneumonia caused by SARS-CoV-2.

**Figure 2 viruses-14-01588-f002:**
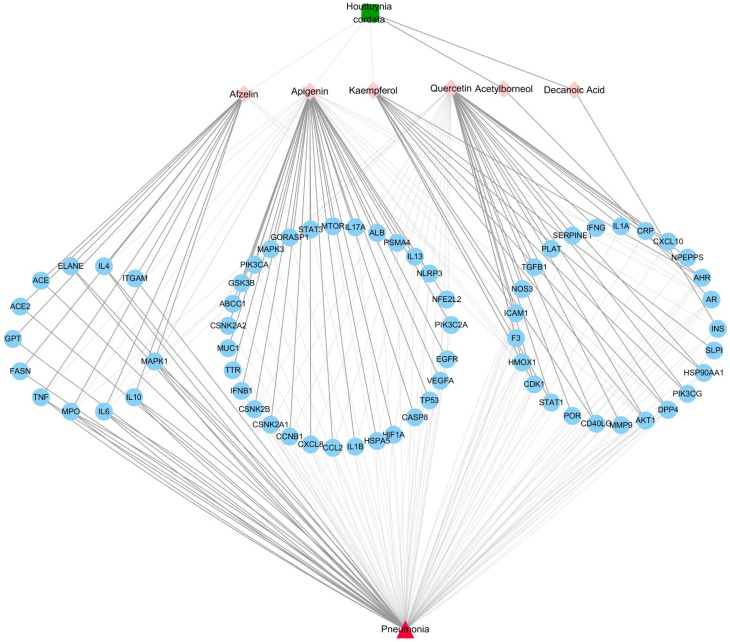
Herb-Metabolites-Targets-Disease network (HMTD) of *Houttuynia cordata*. The green square represents the herb *Houttuynia cordata*; pink diamonds are the active ingredients of *Houttuynia cordata*; blue circles are common targets resulting from the intersection of herb targets and disease targets; the red triangle is the disease.

**Figure 3 viruses-14-01588-f003:**
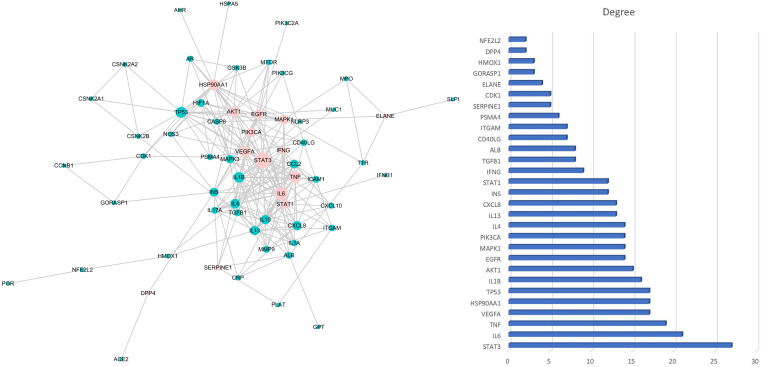
Protein–Protein Interaction network (PPI) of common targets. The size of nodes in the left picture represents the degree of targets. The picture on the right-hand side is the key protein in the PPI network and correlated degree. The y-axis is the gene symbol of the key targets, and the x-axis is the degree of the targets.

**Figure 4 viruses-14-01588-f004:**
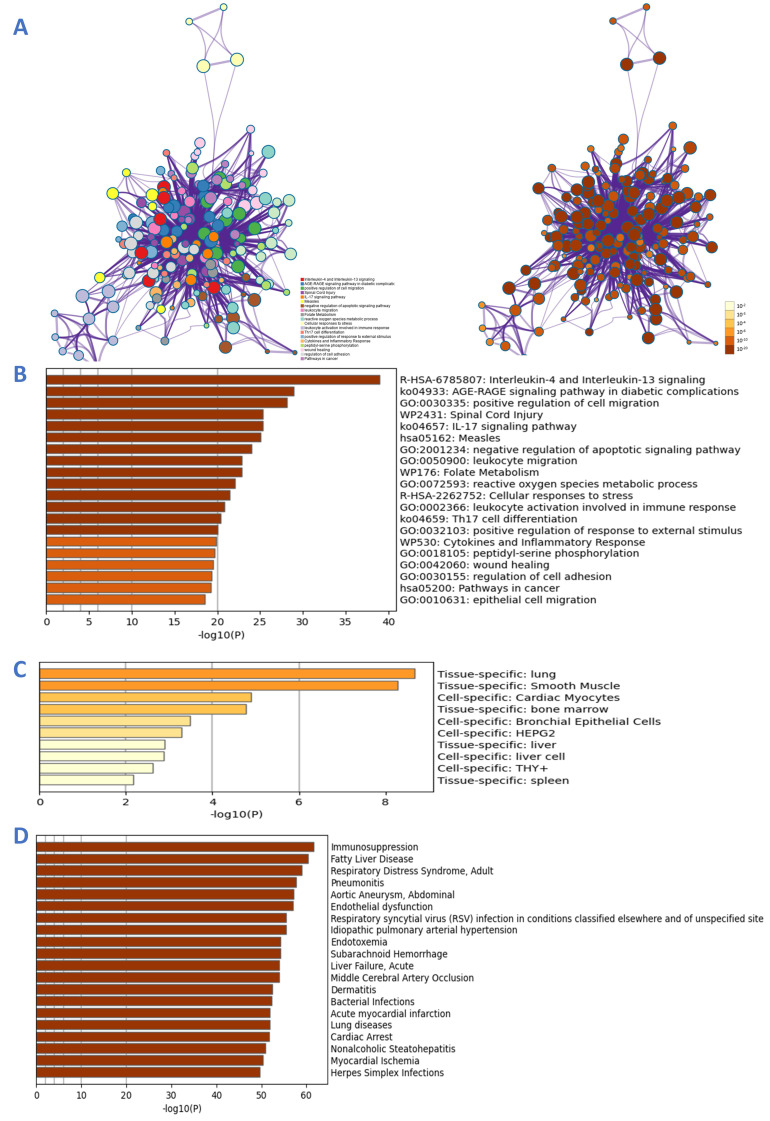
GO analyses of the 67 common targets associated with pneumonia. (**A**) Network of enriched terms: the left-hand side is colored by cluster ID with the best *p*-values from each of 20 clusters wherein nodes that share the same cluster-ID are typically close to each other; the right-hand side network was colored by *p*-value, indicating that terms containing more genes tend to have a more significant *p*-value; (**B**) bar graph of enriched terms colored by *p*-values to visualize the top 20 clusters; (**C**) summary of enrichment analysis in PaGenBase to demonstrate disease targets–organs location; (**D**) summary of enrichment analysis in DisGeNET to show the relevant diseases.

**Figure 5 viruses-14-01588-f005:**
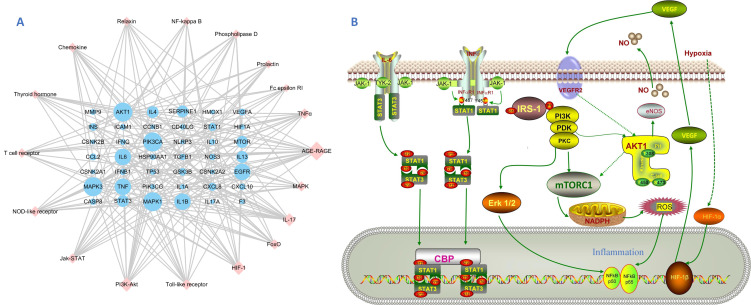
Key pathway network construction based on KEGG enrichment. (**A**) Gene–pathway of *Houttuynia cordata* against pneumonia. The node’s size is related to a degree; pink diamonds are signaling pathways, and blue circles are target genes. (**B**) Schematic diagram of key signaling pathways of *Houttuynia cordata* in treating pneumonia. Solid lines are direct actions, and dashed lines are indirect actions. Some intermediate molecules are not presented. Detailed information and key genes are listed in Table 3.

**Figure 6 viruses-14-01588-f006:**
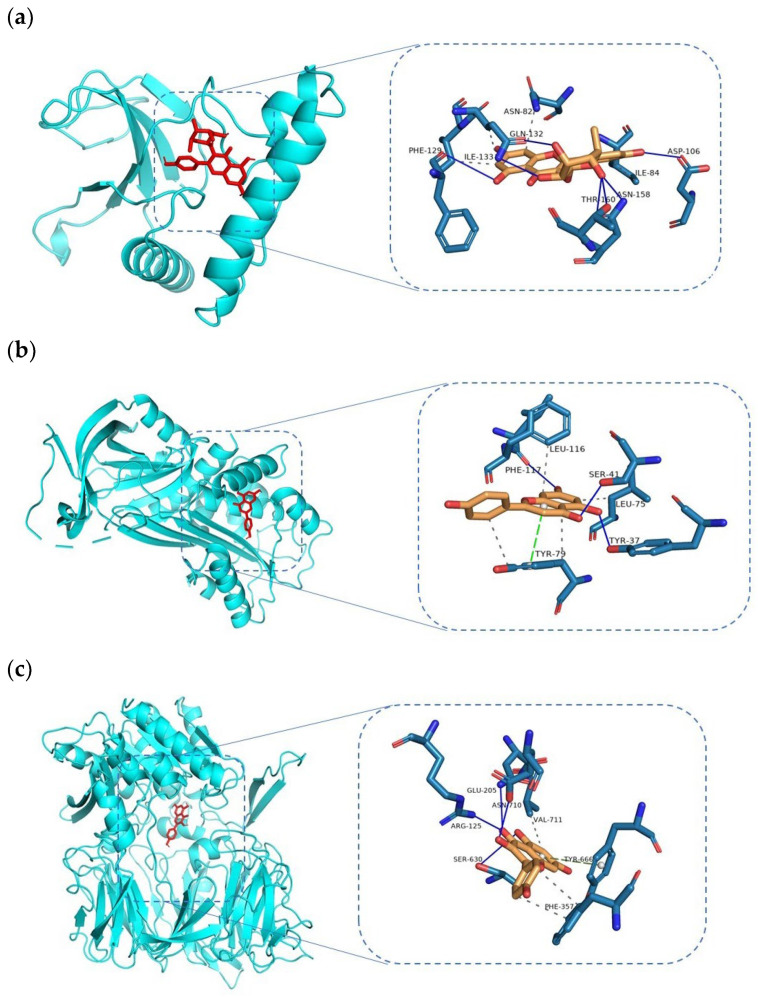
Docking patterns of selected key targets according to the lowest binding affinities with each active metabolite. The TARGET–metabolite complexes include (**a**) MAPK1–afzelin, (**b**) SERPINE1–apigenin, (**c**) DPP4–kaempferol, and (**d**) SERPINE1–apigenin. The binding affinities (kcal/mol) and binding residues are listed in Table 4.

**Figure 7 viruses-14-01588-f007:**
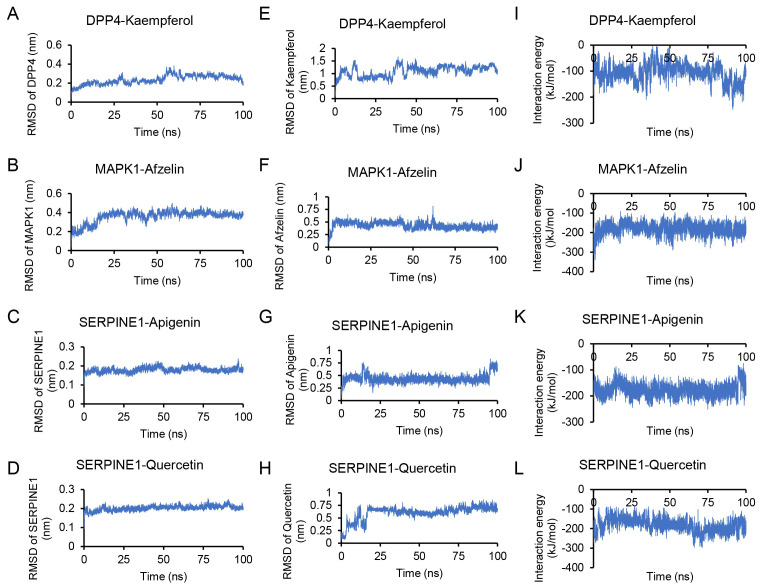
Molecular dynamics simulation results. (**A**–**C**) Molecular dynamics simulation results of DPP4–kaempferol. Root square deviation (RMSD) of DPP4 skeleton atom (**A**), the RMSD of kaempferol heavy atom (**B**), and the interaction energy between DPP4 and kaempferol during 100 ns simulation (**C**). (**D**–**F**) Molecular dynamics simulation results of MAPK1–afzelin. RMSD of MAPK1 skeleton atom (D), the RMSD of afzelin heavy atom (**E**), and the interaction energy between MAPK1 and afzelin during 100 ns simulation (**F**). (**G**–**I**) Molecular dynamics simulation results of SERPINE1–apigenin. RMSD of SERPINE1 skeleton atom (**G**), the RMSD of apigenin heavy atom (**H**), and the interaction energy between SERPINE1 and apigenin during 100 ns simulation (**I**). (**J**–**L**) Molecular dynamics simulation results of SERPINE1–quercetin. RMSD of SERPINE1 skeleton atom (**J**), the RMSD of quercetin heavy atom (**K**) and the interaction energy between SERPINE1 and quercetin during 100 ns simulation (**L**).

**Table 1 viruses-14-01588-t001:** Core metabolites of *Houttuynia cordata*.

PubChem CID	Name	OB (%)	DL	Structure
5280343	Quercetin	46.4	0.27	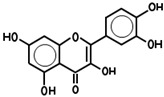
5280459	Quercitrin	4.03	0.73	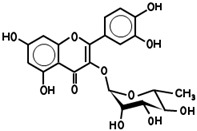
5280863	Kaempferol	41.8	0.24	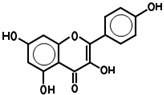
2969	Decanoic Acid	26.74	0.03	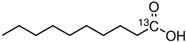
73083158	Acetylborneol		0.63	N/A
5316673	Afzelin	3.83	0.69	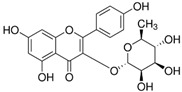
5280443	Apigenin	23.06	0.21	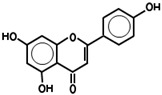

**Table 2 viruses-14-01588-t002:** Information of 21 core genes after protein–protein network analysis.

No	UniProt ID	Gene Symbol	Protein Name	Degree
1	P10275	AR	Androgen receptor	8
2	P31749	AKT1	RAC-alpha serine	8
3	P09601	HMOX1	Heme oxygenase 1	8
4	P27487	DPP4	Dipeptidyl peptidase 4	6
5	P06493	CDK1	Cyclin-dependent kinase 1	6
6	P05362	ICAM1	Intercellular adhesion molecule 1	6
7	P35869	AHR	Aryl hydrocarbon receptor	6
8	P28482	MAPK1	Mitogen-activated protein kinase 1	6
9	P22301	IL10	Interleukin-10	6
10	P05231	IL6	Interleukin-6	6
11	P05164	MPO	Myeloperoxidase	6
12	P01308	INS	Insulin	4
13	P07900	HSP90AA1	Heat shock protein HSP 90-alpha	4
14	P48736	PIK3CG	Phosphatidylinositol 4,5-bisphosphate 3-kinase catalytic subunit gamma isoform	4
15	P29965	CD40LG	CD40 ligand	4
16	P42224	STAT1	Signal transducer and activator of transcription 1-alpha/beta	4
17	P01375	TNF	Tumour necrosis factor	4
18	P08246	ELANE	Neutrophil elastase	4
19	P05112	IL4	Interleukin-4	4
20	P00533	EGFR	Epidermal growth factor receptor	4
21	P15692	VEGFA	Vascular endothelial growth factor A	4

**Table 3 viruses-14-01588-t003:** Virus-related signaling pathway enriched by target genes.

ID	Description	Count	Gene Ratio	FDR
hsa04933	AGE-RAGE signaling pathway in diabetic complications	18	18/67	2.78 × 10^−23^
hsa04657	IL-17 signaling pathway	16	16/67	1.91 × 10^−20^
hsa04066	HIF-1 signaling pathway	15	15/67	1.01 × 10^−18^
hsa04620	Toll-like receptor signaling pathway	12	12/67	4.25 × 10^−14^
hsa04668	TNF signaling pathway	12	12/67	6.79 × 10^−14^
hsa04151	PI3K-Akt signaling pathway	16	16/67	5.19 × 10^−13^
hsa04630	Jak-STAT signaling pathway	12	12/67	3.54 × 10^−12^
hsa04621	NOD-like receptor signaling pathway	12	12/67	5.17 × 10^−12^
hsa04068	FoxO signaling pathway	11	11/67	9.46 × 10^−12^
hsa04660	T cell receptor signaling pathway	10	10/67	1.98 × 10^−11^
hsa04917	Prolactin signaling pathway	8	8/67	9.06 × 10^−10^
hsa04919	Thyroid hormone signaling pathway	9	9/67	1.51 × 10^−9^
hsa04062	Chemokine signaling pathway	10	10/67	3.64 × 10^−9^
hsa04926	Relaxin signaling pathway	9	9/67	4.03 × 10^−9^
hsa04064	NF-kB signaling pathway	8	8/67	7.15 × 10^−9^
hsa04072	Phospholipase D signaling pathway	9	9/67	9.09 × 10^−9^
hsa04664	Fc epsilon RI signaling pathway	7	7/67	1.90 × 10^−8^
hsa04010	MAPK signaling pathway	11	11/67	1.94 × 10^−8^

**Table 4 viruses-14-01588-t004:** Binding affinities (kcal/mol) and binding residues.

Ingredient–Target	BindingAffinity	Binding Residues		
		H-Bonds	Hydrophobic Interaction	π-Stacking/Salt Bridge
Afzelin–IL6	−6.7	TYR-97, ASN-63, THR-137	ASP-140, GLU-93	N/A
Afzelin–DPP4	−8.6	ARG-560, TYR-631, GLY-632, TRP-629, TYR-547, LYS-554	N/A	VAL-546, ASP-545
Afzelin–ELANE	−6.7	ARG-23, CYS-136, GLN-122, GLY-207	PHE-29, LEU-137, TRP-27	N/A
Afzelin–MAPK1	-9.4	PHE-129, GLN-132, ASP-106, ILE-84, ASN-158, THR-150	ILE-133, ASN-82	N/A
Afzelin–HSP90AA1	−7.6	GLN-133	N/A	ARG-46
Afzelin–SERPINE1	−8.1	SER-119, ASP-95, THR-94, TYR-79	PHE-117, ARG-76	N/A
Apigenin–IL6	−6.7	GLN-152, ASN-103, ARG-104, ASP-160	GLN-159, GLN-156	N/A
Apigenin–DPP4	−8.4	SER-630, VAL-546, TYR-547	TRP-629	N/A
Apigenin–ELANE	−7.6	CYS-168, ARG-178	PRO-230, AL-181, THR-164	N/A
Apigenin–MAPK1	−8.1	ILE-133, ASN-154, GLN-132	LEU-150, ILE-140, LEU-155, LEU-157	N/A
Apigenin–HSP90AA1	−7.3	LYS-58, PHE-138, GLY-135	THR-184, LEU-107, THR-109, ASN-51	N/A
Apigenin–SERPINE1	−8.5	PHE-117, SER-41, TYR-37, TYR-39	LEU-116, LEU-75, TYR-79	N/A
Kaempferol–IL6	−6.8	GLN-156, GLN-159, ARG-104	GLN-152	N/A
Kaempferol–DPP4	−8.1	GLU-205, ASN-710, ARG-125, SER-630,	VAL-711, PHE-357	TYR-666
Kaempferol–ELANE	−7.2	ASN-180, THR-164	VAL-181, LEU-130	N/A
Kaempferol–MAPK1	−8.3	GLN-132, LEU-156	LEU-157, ILE-140, LEU-150	N/A
Kaempferol–HSP90AA1	−7.4	LYS-58, ASN-51	THR-109, LEU-107, PHE-138, THR-184	N/A
Kaempferol–SERPINE1	−8.5	ASP-95, PHE-117, LEU-75, ALA-72	SER-41, TYR-79	N/A
Quercetin–IL6	−7.2	GLN-152, ARG-104	GLN-156, GLN-159	N/A
Quercetin–DPP4	−8.5	SER-630, TYR-662, ASN-710, ARG-125, ARG-358	TYR-666, PHE-357	N/A
Quercetin–ELANE	−7.3	ARG-128, CYS-168, GLN-233, ARG-129, ARG-176	THR-164, LEU-130, VAL-181	N/A
Quercetin–MAPK1	−8.5	HIS-147, ASN-82	LEU-155, LEU-156, LEU-157	N/A
Quercetin–HSP90AA1	−7.4	GLY-97, THR-184, LEU-107, GLY-135	ALA-55, ASN-51, ASP-54	N/A
Quercetin–SERPINE1	−8.7	SER-41, ASP-95, SER-119, TYR-37, LEU-75, PHE-117	TYR-79, LEU-116	N/A

## Data Availability

The raw data of this paper are available on request.

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
