# Peer review of "Network Pharmacology and Molecular Docking Elucidate the Underlying Pharmacological Mechanisms of the Herb Houttuynia cordata in Treating Pneumonia Caused by SARS-CoV-2"

_viruses, 2022, doi:10.3390/v14071588_

Round 1

Reviewer 1 Report

Comments for the manuscript entitled “Network pharmacology and molecular docking elucidate the 2 underlying pharmacological mechanisms of the herb 3 Houttuynia cordata in treating pneumonia caused by 4 SARS-CoV-2” prepared for Viruses.

The authors have analyzed the mechanisms of Houttuynia cordata metabolites against SARS-CoV-2 by mainly focusing on molecular network analyses and molecular modeling. The authors proposed four Houttuynia cordata metabolites and six human target proteins. The attempt of this study, elucidating the drug mechanisms of natural compounds through analyses of molecular interaction network, is interesting, and the topic is suitable for this journal. Thus, this reviewer is basically positive to recommend this article for publication if the points raised in the attached comments are appropriately addressed in the revision process.

Major points

1)      Lines 116-120: Identifying human targets for natural compounds is one of the most critical steps in this kind of study. In the presented study, the authors used “Traditional Chinese Medicine Systems Pharmacology Database and Analysis Platform (TCMSP), the TCM Integrated Database (TCMID), The Encyclopaedia of Traditional Chinese Medicine (ETCM), and the Bioinformatics Analysis Tool for Molecular Mechanism of TCM (BATMAN-TCM) [14]” for this purpose. These databases are not popular ones, and their reliabilities are unclear. The reference (Ref. 14) is the report of another study of this kind, and thus it is not appropriate. Also the significance of the employed criteria “DL>0.18 and OB>30%” and “score>0.8” is not explained at all. Thus, (i) The authors must refer to the original literatures for these databases, and properly explain the reliability of these database. (ii) The authors must quantitatively evaluate the accuracy of the predictions by using these databases.

2)      Lines 384-385 “The binding affinities of all docking patterns were less than -6 kcal/mol, indicating a stable binding between active ingredients and protein targets.”: Indicating negative dG value does not generally prove stable binding. The dG values under same simulation parameters for several control cases, i.e., of compounds for which bindings or non-bindings were experimentally demonstrated must be shown.

3)      Lines 210-211 “mainly within the areas of docking pockets, i.e., active pocket sites where small-molecule ligands bind.” Identification of ligand-binding pocket on protein structure is not a minor problem. Some quantitative criteria for defining pockets should be explained.

4)      Lines 221-222 “The system was neutralized by Na+ and Cl-, followed by adding 0.15 M NaCl.”; The system would not be neutralized thorough this procedure. Is this actually intended? If so, why must this be done?

Minor points:

5)      Line 23: The abbreviation “HC” is not defined.

6)      Lines 45-47 “This herb has traditionally been used … to have a preventive effect on SARS-CoV-1 infection.” Show reference for this.

7)      Fix the inconsistent notations like “NF-kB” and “NF-kappa-B” or “p-value” and “P-value” throughput the manuscript.

8)      Line 156-157 “Homo Sapience” “Homo sapience”.

9)      Lines 207-109 “The key target proteins were purposefully selected with a ligand-binding model smaller than 2”: How a model can be smaller than ‘2’?

10)    Lines 236-238 “A number of metabolites had already been established as the most effective components of HC throughout the relevant literature.” Show the references for this.

11)    Lines 289-290 “The picture on the right-hand side is the key protein in the PPI network and correlated degree.” What is “correlated degree”? Does this sentence mean “is the key proteins … sorted by degree”?

12)    Lines 322-323 “the same cluster-ID are typically close to each other; (b) Coloured 322 by p-value”: Closeness of nodes belonging to a same cluster (same color?) is not clear in this Figure because edges are not resolved. Also, there is no such plate (b) in this Figure.

13)    Lines 330-331 “Schematic diagram of key signalling path-ways of Houttuynia cordata in treating pneumonia.”: Where did this diagram come from? It looks being copied without citation.

14)    Lines 377-387: Scales for displacements should be aligned among plates A-D and plates E-H in Figure 7.

15)    Sections are too long without break in the Discussion part.

Author Response

Question 1)      Lines 116-120: Identifying human targets for natural compounds is one of the most critical steps in this kind of study. In the presented study, the authors used “Traditional Chinese Medicine Systems Pharmacology Database and Analysis Platform (TCMSP), the TCM Integrated Database (TCMID), The Encyclopaedia of Traditional Chinese Medicine (ETCM), and the Bioinformatics Analysis Tool for Molecular Mechanism of TCM (BATMAN-TCM) [14]” for this purpose. These databases are not popular ones, and their reliabilities are unclear. The reference (Ref. 14) is the report of another study of this kind, and thus it is not appropriate. Also the significance of the employed criteria “DL>0.18 and OB>30%” and “score>0.8” is not explained at all. Thus, (i) The authors must refer to the original literatures for these databases, and properly explain the reliability of these database. (ii) The authors must quantitatively evaluate the accuracy of the predictions by using these databases.

Response: Thank you very much for the constructive comments. Regarding your questions: 1) the reference of these databases was from these book Network Pharmacology edited by Shao Li. See the development of network pharmacology.

Taking ETCM as an example, this database employs the MedChem Studio (version 3.0, commercial software) to predict the potential targets of TCM ingredients. MedChem Studio, a drug similarity search tool, is utilized to find known drugs that possess high structural similarity (Tanimoto>0.8) with TCM ingredients, facilitating target prediction. MedChem Studio was developed by SimulationPlus, which is headquartered in Lancaster, CA, but has employees on both the East and West Coasts of the United States, Canada, Europe and India. Our staff is multidisciplinary, including backgrounds in machine learning, physiologically based biopharmaceutics/pharmacokinetics, toxicology, and population PK/PD modeling approaches. We work in cross-functional teams to develop ADMET, PBBM, and PBPK platforms to reduce the costs of R&D through innovative science-based software and consulting solutions that optimize treatment options and improve patient lives. Please click on their photos below to read more about individual team members; 2) Regarding to the accuracy of the predictions by using these databases, network pharmacology aims to analyze the mechanism of clinically effective drugs and prescriptions, and provides an understanding and interpretation of pharmacological mechanisms and a new record for drug-target relationship data resources. This system strives to merge the knowledge accumulated over 2000 years of clinical practices with the modern experiences or calculation methods. HC is a traditional herbal medicine recognized for its favorable antiviral properties, particularly against clinically enveloped viruses which has found use in Asian countries, including China, Japan, and Thailand. We think that network pharmacology and these databases can elucidate the mechanisms of its pharmacological activities and investigate its therapeutic potential.

Question 2)      Lines 384-385 “The binding affinities of all docking patterns were less than -6 kcal/mol, indicating a stable binding between active ingredients and protein targets.”: Indicating negative dG value does not generally prove stable binding. The dG values under same simulation parameters for several control cases, i.e., of compounds for which bindings or non-bindings were experimentally demonstrated must be shown.

Response: Yes, you are correct, the dg value doesn’t generally prove stable binding, that’s why we introduced other methods for judgment, such as molecular dynamics simulation. Molecular dynamics (MD) simulation is the most appropriate tool to quantify the molecular structure and binding/adsorption free energies. Our MD simulations showed that RMSD of proteins and ingredients in the complex remained stable during the simulation and the interaction energy remained at a very low level, indicating that the combination of the proteins and molecules was stable in the simulation (Figure 7). Accordingly, our studies provide a reference for subsequent experimental design.

Question 3)      Lines 210-211 “mainly within the areas of docking pockets, i.e., active pocket sites where small-molecule ligands bind.” Identification of ligand-binding pocket on protein structure is not a minor problem. Some quantitative criteria for defining pockets should be explained.

Response: Thank you for your recommendation. To make it clearer, we have added the following statement and related references in the revised manuscript (lines 212-214): “The active site of the protein is centered on the active amino acid site of the original ligand in the crystal structure, which residue information can be obtained from the literature [29-34].”

  1. Namoto, K.; Sirockin, F.; Ostermann, N.; Gessier, F.; Flohr, S.; Sedrani, R.; Gerhartz, B.; Trappe, J.; Hassiepen, U.; Duttaroy, A.; Ferreira, S.; Sutton, J. M.; Clark, D. E.; Fenton, G.; Beswick, M.; Baeschlin, D. K., Discovery of C-(1-aryl-cyclohexyl)-methylamines as selective, orally available inhibitors of dipeptidyl peptidase IV. Bioorg Med Chem Lett 2014, 24, (3), 731-6.
  2. Lechtenberg, B. C.; Kasperkiewicz, P.; Robinson, H.; Drag, M.; Riedl, S. J., The elastase-PK101 structure: mechanism of an ultrasensitive activity-based probe revealed. ACS Chem Biol 2015, 10, (4), 945-51.
  3. Amaral, M.; Kokh, D. B.; Bomke, J.; Wegener, A.; Buchstaller, H. P.; Eggenweiler, H. M.; Matias, P.; Sirrenberg, C.; Wade, R. C.; Frech, M., Protein conformational flexibility modulates kinetics and thermodynamics of drug binding. Nat Commun 2017, 8, (1), 2276.
  4. Shaw, S.; Bourne, T.; Meier, C.; Carrington, B.; Gelinas, R.; Henry, A.; Popplewell, A.; Adams, R.; Baker, T.; Rapecki, S.; Marshall, D.; Moore, A.; Neale, H.; Lawson, A., Discovery and characterization of olokizumab: a humanized antibody targeting interleukin-6 and neutralizing gp130-signaling. MAbs 2014, 6, (3), 774-82.
  5. O'Reilly, M.; Cleasby, A.; Davies, T. G.; Hall, R. J.; Ludlow, R. F.; Murray, C. W.; Tisi, D.; Jhoti, H., Crystallographic screening using ultra-low-molecular-weight ligands to guide drug design. Drug Discov Today 2019, 24, (5), 1081-1086.
  6. Sillen, M.; Miyata, T.; Vaughan, D. E.; Strelkov, S. V.; Declerck, P. J., Structural Insight into the Two-Step Mechanism of PAI-1 Inhibition by Small Molecule TM5484. Int J Mol Sci 2021, 22, (3).

Question 4)      Lines 221-222 “The system was neutralized by Na+ and Cl-, followed by adding 0.15 M NaCl.”; The system would not be neutralized thorough this procedure. Is this actually intended? If so, why must this be done?

Response: Thank you so much for your valuable comment. 0.15 M NaCl is the concentration of normal saline. Neutralizing charge and adding 0.15 M NaCl in the system is to make the system close to physiological state. We added the reason to the method (lines 225-226). In addition, molecular dynamics simulation software GROMACS uses Particle-Mesh-Ewald electrostatic algorithm, which is inaccurate for systems where charge is not conserved. Therefore, the charge of the system must be neutralized.

Question 5)      Line 23: The abbreviation “HC” is not defined.

Response: It has been defined now.

Question 6)      Lines 45-47 “This herb has traditionally been used … to have a preventive effect on SARS-CoV-1 infection.” Show reference for this.

Response: The reference has been put in the text.

Lau KM, Lee KM, Koon CM, Cheung CS, Lau CP, Ho HM, Lee MY, Au SW, Cheng CH, Lau CB, Tsui SK, Wan DC, Waye MM, Wong KB, Wong CK, Lam CW, Leung PC, Fung KP. Immunomodulatory and anti-SARS activities of Houttuynia cordata. J Ethnopharmacol. 2008 Jun 19;118(1):79-85. doi: 10.1016/j.jep.2008.03.018.

Question 7)      Fix the inconsistent notations like “NF-kB” and “NF-kappa-B” or “p-value” and “P-value” throughput the manuscript.

Response: We have modified them accordingly.

Question 8)      Line 156-157 “Homo Sapience” à “Homo sapience”.

Response: We have revised it.

Question 9)      Lines 207-109 “The key target proteins were purposefully selected with a ligand-binding model smaller than 2”: How a model can be smaller than ‘2’?

Response: It should be “The key target proteins were purposefully selected with a resolution smaller than 2, and their crystals were imported into PyMOL 3.0 software” (lines 210-211).

Question 10)    Lines 236-238 “A number of metabolites had already been established as the most effective components of HC throughout the relevant literature.” Show the references for this.

Response: The relevant literature has been added to the text.

Question 11)    Lines 289-290 “The picture on the right-hand side is the key protein in the PPI network and correlated degree.” What is “correlated degree”? Does this sentence mean “is the key proteins … sorted by degree”?

Response: The main topological parameters including, betweenness centrality, closeness centrality, and degree of the PPI network. The hub genes in the network were based on degree values. Degree shows the number of connected nodes with the individual node. Therefore, higher degree indicates a characteristic of hub.

Question 12)    Lines 322-323 “the same cluster-ID are typically close to each other; (b) Coloured 322 by p-value”: Closeness of nodes belonging to a same cluster (same color?) is not clear in this Figure because edges are not resolved. Also, there is no such plate (b) in this Figure.

Response: We have modified it. As there is no b plate, it should be right-hand side plate in A. The same enrichment network has its nodes colored by p-value, as shown in the legend. The dark the color, the more statistically significant the node is (see legend for p-value ranges).

Question 13)    Lines 330-331 “Schematic diagram of key signalling path-ways of Houttuynia cordata in treating pneumonia.”: Where did this diagram come from? It looks being copied without citation.

Response: We drew this graph by integrating our results in one picture, not copying it from any publication.

Question 14)    Lines 377-387: Scales for displacements should be aligned among plates A-D and plates E-H in Figure 7.

Response:Thank you for pointing this out. We have corrected the error accordingly (Figure 7).

Question 15)    Sections are too long without break in the Discussion part.

 Response: Thanks, we have modified them accordingly.

Reviewer 2 Report

This manuscript mainly carried out pharmacological mechanism of Houttuynia cordata on pneumonia caused by SARs-Cov-2 using network pharmacology and molecular docking. The bioinformatics analysis was performed to predict this pharmacological mechanism, showing differentially expressed genes by public RNA-Seq and cDNA microarray data, drug-target interaction network, protein-protein interaction prediction, gene set enrichment test, and docking simulation between the compounds of Houttuynia cordata and target candidate proteins. Finally, the manuscirpt suggested the potential mechanisms of HC in pneumonia.

The manuscript is well-written and the experiments are well designed and the results are reasonably discussed. However, some minor points should be addressed by the authors.

 1. More details are required for methods. For instance, reference is necessary when stating the definition for betweenness. The exact section explains the terms closeness, centrality, and degrees with appropriate references.

 2. Please provide the importance of using these criteria for selecting/shortlisting your compounds. For the screening of ingredients and OB and DL, many other properties also affect the drug activity, such as water solubility, oil-water partition coefficient, plasma protein binding rate, P-glycoprotein, P450 enzyme metabolites, drug half-life, and blood components. For example, are the seven ingredients screened at the end of the article the ingredients with higher content in Houttuynia cordata?

 3. Please refrain from suggesting any results in the method section. Instead, add all results in the result section.

 4. Were any positive controls used in the study?

 5. Please justify why you used specific tools and why not other similar tools for carrying out your work.

 6. For the discussion, please provide more discussion on the possibilities of the outcomes you have obtained, a comparison between the results in your work and other similar work, and more details if the scientific hypothesis initially postulated in the study was satisfied from the outcomes of your work or not.

 7. In terms of molecular docking, the residue composition of the active sites of different targets is different, and the docking scores are energy-based, so the docking scores of other targets cannot be compared directly; the compounds with the top docking scores can bind better to the targets, and this standard-setting is too subjective and unreasonable.

Author Response

  1. More details are required for methods. For instance, reference is necessary when stating the definition for betweenness. The exact section explains the terms closeness, centrality, and degrees with appropriate references.

Response: We have modified the text with more accurate definitions of these measures.

  1. Please provide the importance of using these criteria for selecting/shortlisting your compounds. For the screening of ingredients and OB and DL, many other properties also affect the drug activity, such as water solubility, oil-water partition coefficient, plasma protein binding rate, P-glycoprotein, P450 enzyme metabolites, drug half-life, and blood components. For example, are the seven ingredients screened at the end of the article the ingredients with higher content in Houttuynia cordata?

Response: Yes, many properties can affect the drug activity, which may requires the further validation. This manuscript tried to provide a theoretical basis for further study on Houttuynia's active drug-like ingredients and mechanism in pneumonia treatment. Two factors of ADME were considered according to the databases suggestions: 1) the related compounds were comprehensively studied by using a few compounds as possible; 2) the reported pharmacological data was used to explain and establish the model more reasonably. Therefore, the screening candidate compounds (seven) with OB>26% and DL>0.18 can be used for subsequent studies.

  1. Please refrain from suggesting any results in the method section. Instead, add all results in the result section.

Response: We have tried to refrain from putting the results in the method section. However, since each step of analysis procedure was linked to each other, we tried to make each step understandable.

  1. Were any positive controls used in the study?

Response: No, this is a virtual screening of drug candidates, thus being no positive controls.

  1. Please justify why you used specific tools and why not other similar tools for carrying out your work.

Response: We used these tools suggested by the book Network Pharmacology because they are freely available and robust.

  1. For the discussion, please provide more discussion on the possibilities of the outcomes you have obtained, a comparison between the results in your work and other similar work, and more details if the scientific hypothesis initially postulated in the study was satisfied from the outcomes of your work or not.

Response: Based on the data on Houttuynia cordata, we believe that the plant exerts collective therapeutic effects against pneumonia, and thus provides a theoretical basis for further study of the active drug-like ingredients and mechanism of Houttuynia cordata in the treatment of pneumonia. This study confirms the potential therapeutic effects of Houttuynia cordata, a time-honored herb widely used in Asian countries to treat pneumonia and provides new data on ancient therapy.

  1. In terms of molecular docking, the residue composition of the active sites of different targets is different, and the docking scores are energy-based, so the docking scores of other targets cannot be compared directly; the compounds with the top docking scores can bind better to the targets, and this standard-setting is too subjective and unreasonable.

Response: Thank you for pointing this out. We used docking to initially judge the possibility of compounds binding to different proteins. The same compound is not comparable with different proteins. To further confirm the possibility of compounds binding to proteins, molecular dynamics simulations were used to explore the stability of protein and compound structures during the simulation process and calculate the interaction energy between them.

Round 2

Reviewer 1 Report

The manuscript was revised almost accordingly to the comments of this reviewer except for “Question 12)  Closeness of nodes belonging to a same cluster (same color?) is not clear in this Figure because edges are not resolved.”, for which the authors did nothing to revised the Figure. If this point was appropriately fixed, this reviewer would recommend publication of this paper.

Author Response

Reviewer 1’s comment

The manuscript was revised almost accordingly to the comments of this reviewer except for “Question 12)  Closeness of nodes belonging to a same cluster (same color?) is not clear in this Figure because edges are not resolved.”, for which the authors did nothing to revised the Figure. If this point was appropriately fixed, this reviewer would recommend publication of this paper.

Response: Thank you very much for your new comment. We are sorry for the unclear explanation in our previous responses. There is no need to modify the graph in the Figure 4A. Since there is no b plate, it should be right-hand side plate in A. The same enrichment network has its nodes colored by p-value, as shown in the legend. The dark the color, the more statistically significant the node is (see legend for p-value ranges).

Actually, we did modify the legend to make it clearer and more explainable in our previous Response as follows: We changed the figure legend from “Figure 4. GO analyses of the 67 common targets associated with pneumonia. A) Network of enriched terms: the left-hand side is coloured by cluster ID with the best p-values from each of 20 clusters wherein nodes that share the same cluster-ID are typically close to each other;” to “Figure 4. GO analyses of the 67 common targets associated with pneumonia. A) Network of enriched terms: the one on the left-hand side is coloured by cluster ID with the best p-values from each of 20 clusters wherein nodes that share the same cluster-ID are typically close to each other; the one on the right-hand side network was coloured by p-value, indicating that terms containing more genes tend to have a more significant p-value;”
